# RAPID: RETRIEVAL AUGMENTED TRAINING OF DIFFERENTIALLY PRIVATE DIFFUSION MODELS

**Tanqiu Jiang**[†]   **Changjiang Li**[†]   **Fenglong Ma**[⋆]   **Ting Wang**[†]
[†]Stony Brook University   ⋆Pennsylvania State University

## ABSTRACT

Differentially private diffusion models (DPDMs) harness the remarkable generative capabilities of diffusion models while enforcing differential privacy (DP) for sensitive data. However, existing DPDM training approaches often suffer from significant utility loss, large memory footprint, and expensive inference cost, impeding their practical uses. To overcome such limitations, we present RAPID[1], a novel approach that integrates retrieval augmented generation (RAG) into DPDM training. Specifically, RAPID leverages available public data to build a knowledge base of sample trajectories; when training the diffusion model on private data, RAPID computes the early sampling steps as queries, retrieves similar trajectories from the knowledge base as surrogates, and focuses on training the later sampling steps in a differentially private manner. Extensive evaluation using benchmark datasets and models demonstrates that, with the same privacy guarantee, RAPID significantly outperforms state-of-the-art approaches by large margins in generative quality, memory footprint, and inference cost, suggesting that retrieval-augmented DP training represents a promising direction for developing future privacy-preserving generative models. The code is available at: https://github.com/TanqiuJiang/RAPID.

## 1 INTRODUCTION

The recent advances in diffusion models have led to unprecedented capabilities of generating high-quality, multi-modal data (Ho et al., 2020; Kong et al., 2020; Bar-Tal et al., 2024). However, training performant diffusion models often requires massive amounts of training data, raising severe privacy concerns in domains wherein data is sensitive. For instance, Carlini et al. (2023) show that compared with other generative models, diffusion models are especially vulnerable to membership inference attacks, due to their remarkable modeling capabilities; meanwhile, Wen et al. (2023) show that text-conditional diffusion models trained with text-image pairs can produce images almost identical to certain training samples with proper prompting.

This pressing need has spurred intensive research on enforcing privacy protection in diffusion model training. Notably, Dockhorn et al. (2023) proposed the concept of differentially private diffusion models (DPDMs), which incorporate DP-SGD (Abadi et al., 2016) into diffusion model training, providing guaranteed privacy; Ghalebikesabi et al. (2023) further applied DP during diffusion model fine-tuning, first pre-training a denoising diffusion probabilistic model (Ho et al., 2020) on public data and then fine-tuning the model on private data under DP constraints. Similarly, Lyu et al. (2023) employed the same strategy but extended it to latent diffusion models (Rombach et al., 2022). However, existing methods suffer from major limitations. (*i*) Significant utility loss – The quality of generated samples often drops sharply under tightened privacy budgets. For instance, DPDM (Dockhorn et al., 2023) fails to synthesize recognizable images on CIFAR10 under a DP budget of $\epsilon = 1$. (*ii*) Large memory footprint – To reduce the noise magnitude applied at each iteration, most approaches adopt excessive batch sizes (e.g., $B = 8,192$ samples per batch). As common DP frameworks (e.g., OPACUS (Yousefpour et al., 2021)) have peak memory requirements of $\mathcal{O}(B^2)$, this severely limits the size of usable diffusion models. (*iii*) Expensive inference cost – Similar to non-private diffusion models, existing methods require expensive, iterative sampling at inference, impeding them from synthesizing massive amounts of data.

---

[1]RAPID: Retrieval-Augmented PrIvate Diffusion model.

To address such challenges, we present RAPID, a novel approach that integrates retrieval augmented generation (RAG) (Lewis et al., 2020; Blattmann et al., 2022) into DPDM training. Our approach is based on the key observation that small perturbations to the early sampling steps of diffusion models have a limited impact on the overall sampling trajectory (Khalil, 2008). This allows us to reuse previously generated trajectories, if similar to the current one, as effective surrogates to reduce both training and inference costs (Zhang et al., 2023). Leveraging this idea, RAPID utilizes available public data to pre-train a diffusion model and build a knowledge base of sampling trajectories. It further refines the model using private data: it first computes the early sampling steps, retrieves similar trajectories from the knowledge base as surrogates, and focuses on training the later sampling steps under DP constraints. Compared to prior work, RAPID offers several major advantages:

- It achieves a more favorable privacy-utility trade-off by fully utilizing public data and only training the later sampling steps on private data;
- It significantly reduces batch-size requirements by leveraging the retrieved sample trajectories, making the use of large diffusion models feasible;
- It greatly improves inference efficiency by skipping intermediate sampling steps via RAG.

Extensive evaluation using benchmark datasets and models demonstrates that RAPID outperforms state-of-the-art methods by large margins in there key areas: (*i*) generative quality (e.g., improving FID score to 63.2 on CIFAR10 under a DP budget $\epsilon = 1$), (*ii*) memory footprint (e.g., reducing the required batch size to just 64 samples per batch), and (*iii*) inference efficiency (e.g., saving up to 50% of the inference cost). Our findings suggest that integrating RAG into DP training represents a promising direction for developing future privacy-preserving generative models.

## 2    RELATED WORK

**Differentially private data generation.** As an important yet challenging problem, enforcing DP into training a variety of advanced generative models has attracted intensive research effort (Hu et al., 2023), including generative adversarial networks (Goodfellow et al., 2014; Chen et al., 2020a; Yoon et al., 2019), variational autoencoders (Jiang et al., 2022), and customized architectures (Liew et al., 2022; Vinaroz et al., 2022; Harder et al., 2021). For instance, Harder et al. (2023) pre-train perceptual features using public data and fine-tune only data-dependent terms using maximum mean discrepancy under the DP constraint.

**Differentially private diffusion models.** In contrast, the work on privatizing diffusion models is relatively limited. Notably, DPDM (Dockhorn et al., 2023) integrate DP-SGD (Abadi et al., 2016) with a score-based diffusion model (Song et al., 2021); Ghalebikesabi et al. (2023) propose to pre-train a diffusion model with public data and then fine-tune the model using DP-SGD on private data; DP-LDM (Lyu et al., 2023) apply a similar fine-tuning strategy to a latent diffusion model (Rombach et al., 2022); PrivImage (Li et al., 2024) queries the private data distribution to select semantically similar public samples for pretraining, followed by DP-SGD fine-tuning on the private data. However, all these methods share the drawbacks of significant utility loss, excessive batch sizes, or expensive inference costs. Beyond the pre-training/fine-tuning paradigm, recent work also explores synthesizing DP datasets by querying commercial image generation APIs (e.g., Stable Diffusion and DALL-E2) to approximate the distributions of private data (Wang et al., 2024; Lin et al., 2024).

**Retrieval augmented generation.** Initially proposed to enhance the generative quality of NLP models by retrieving related information from external sources Khandelwal et al. (2019); Lewis et al. (2020); Guu et al. (2020), RAG has been extended to utilize local cohorts in the training data to facilitate image synthesis. For instance, rather than directly outputting the synthesized sample, Casanova et al. (2021) compute the average of the sample's nearest neighbors in the training data. Blattmann et al. (2022) use an external image dataset to provide enhanced conditional guidance, augmenting the text prompt during the text-to-image generation. Zhang et al. (2023) explore RAG to accelerate the inference process of a diffusion model by reusing pre-computed sample trajectories as surrogates for skipping intermediate sampling steps. However, existing work primarily focuses on employing RAG in the inference stage to improve generative quality or efficiency.

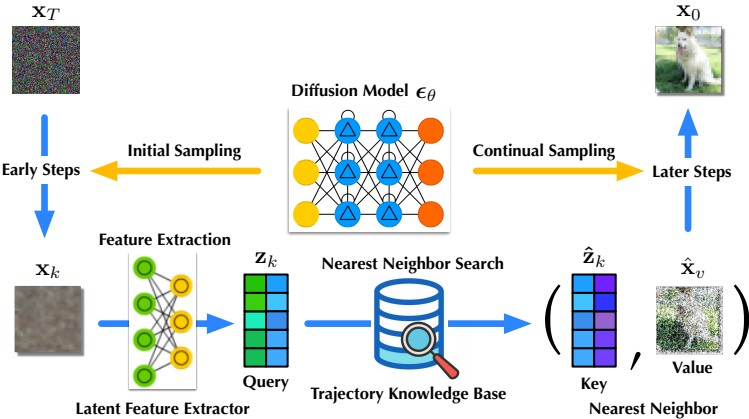

Figure 1: Overall framework of RAPID.

To our best knowledge, this presents the first work on integrating RAG in the DP training of diffusion models, aiming to improve the generative quality, memory footprint, and inference efficiency over the existing DPDM approaches.

## 3 RAPID

Next, we present RAPID, a novel approach for training differentially private diffusion models by leveraging retrieval-augmented generation.

### 3.1 PRELIMINARIES

A diffusion model consists of a forward diffusion process that converts original data $\mathbf{x}_0$ to its latent $\mathbf{x}_t$ (where $t$ denotes the timestep) via progressive noise addition and a reverse sampling process that starts from latent $\mathbf{x}_t$ and generates data $\mathbf{x}_0$ via sequential denoising steps.

Take the denoising diffusion probabilistic model (DDPM) (Ho et al., 2020) as an example. Given $\mathbf{x}_0$ sampled from the real data distribution $q_{\text{data}}$, the diffusion process is formulated as a Markov chain:

$$q(\mathbf{x}_t|\mathbf{x}_{t-1}) = \mathcal{N}(\mathbf{x}_t; \sqrt{1-\beta_t}\mathbf{x}_{t-1}, \beta_t\mathbf{I}) \tag{1}$$

where $\{\beta_t \in (0,1)\}_{t=1}^T$ specifies the variance schedule. For sufficiently large $T$, the latent $\mathbf{x}_T$ approaches an isotropic Gaussian distribution. Starting from $p(\mathbf{x}_T) = \mathcal{N}(\mathbf{x}_T; \mathbf{0}, \mathbf{I})$, the sampling process maps latent $\mathbf{x}_T$ to data $\mathbf{x}_0$ in $q_{\text{data}}$ as a Markov chain with a learned Gaussian transition:

$$p_\theta(\mathbf{x}_{t-1}|\mathbf{x}_t) = \mathcal{N}(\mathbf{x}_{t-1}; \boldsymbol{\mu}_\theta(\mathbf{x}_t, t), \boldsymbol{\Sigma}_\theta(\mathbf{x}_t, t)) \tag{2}$$

To train the diffusion model $\boldsymbol{\epsilon}_\theta(\mathbf{x}_t, t)$ that predicts the cumulative noise up to timestep $t$ for given latent $\mathbf{x}_t$, DDPM aligns the mean of the transition $p_\theta(\mathbf{x}_{t-1}|\mathbf{x}_t)$ with the posterior $q(\mathbf{x}_{t-1}|\mathbf{x}_t, \mathbf{x}_0)$:

$$\min_\theta \mathbb{E}_{\mathbf{x}_0 \sim q_{\text{data}}, t \sim \mathcal{U}(1,T), \boldsymbol{\epsilon} \sim \mathcal{N}(\mathbf{0},\mathbf{I})} \|\boldsymbol{\epsilon} - \boldsymbol{\epsilon}_\theta(\sqrt{\bar{\alpha}_t}\mathbf{x}_0 + \sqrt{1-\bar{\alpha}_t}\boldsymbol{\epsilon}, t)\|^2 \quad \text{where} \quad \bar{\alpha}_t = \prod_{\tau=1}^t (1-\beta_\tau)$$

Once trained, starting from $\mathbf{x}_T \sim \mathcal{N}(\mathbf{0}, \mathbf{I})$, the sampling process iteratively invokes $\boldsymbol{\epsilon}_\theta$:

$$\mathbf{x}_{t-1} = \boldsymbol{\epsilon}_\theta(\mathbf{x}_t, t), \tag{3}$$

which generates the following trajectory $\{\mathbf{x}_T, \mathbf{x}_{T-1}, \dots, \mathbf{x}_0\}$.

### 3.2 DESIGN OF RAPID

Prior work on training DPDMs (Dockhorn et al., 2023; Ghalebikesabi et al., 2023; Lyu et al., 2023) often applies DP-SGD (Abadi et al., 2016) to fine-tune the entire sampling process using private data, resulting in significant utility loss and inference cost. However, it is known that, within a given sampling trajectory $\{\mathbf{x}_T, \mathbf{x}_{T-1}, \dots, \mathbf{x}_0\}$, the early steps only determine the high-level image layout shared by many latents, while the later steps determine the details (Khalil, 2008; Meng et al., 2021).

---

**Algorithm 1:** Training latent feature extractor.

---

**Input:** reference data $\mathcal{D}$, pre-trained diffusion model $\boldsymbol{\epsilon}_\theta$, timestep $k$
**Output:** latent feature extractor $h$

1 **while** *not converged yet* **do**
2     **foreach** $\mathbf{x} \in \mathcal{D}$ **do**
        // generate positive and negative pairs
3         generate $\tilde{\mathbf{x}}$, $\tilde{\mathbf{x}}^+$, and $\mathcal{N}_{\mathbf{x}}^-$ with random augmentations;
        // generate latents at timestep $k$
4         sample $\tilde{\mathbf{x}}_k$, $\tilde{\mathbf{x}}_k^+$, and $\tilde{\mathbf{x}}_k^-$ for $\tilde{\mathbf{x}}^- \in \mathcal{N}_{\mathbf{x}}^-$ following Eq. 3;
        // compute contrastive loss
5         compute $\ell_{\mathrm{CL}}(\mathbf{x})$ following Eq. 4;
        // update feature extractor
6         update $h$ to minimize $\ell_{\mathrm{CL}}(\mathbf{x})$;

7 **return** $h$;

---

Thus, instead of privatizing the end-to-end sampling process, by fully utilizing the public data, we may skip intermediate steps and focus on fine-tuning the later steps using private data.

Motivated by this idea, as illustrated in Figure 1, RAPID first pre-trains a diffusion model $\boldsymbol{\epsilon}_\theta$ using the public data $\mathcal{D}^{\mathrm{pub}}$. Further, RAPID builds a knowledge base $\mathcal{KB}$ by calculating the diffusion trajectories of $\mathcal{D}^{\mathrm{pub}}$. RAPID then fine-tunes $\boldsymbol{\epsilon}_\theta$ using the private data $\mathcal{D}^{\mathrm{prv}}$ as follows. Corresponding to each input $\mathbf{x} \in \mathcal{D}^{\mathrm{prv}}$, it generates its initial steps $\mathbf{x}_{T:k}$ in the sampling process, uses $\mathbf{x}_k$ (at timestep $k$) as a query to retrieve a similar trajectory $\hat{\mathbf{x}}_{T:0}$ from $\mathcal{KB}$, and resumes DP training the sampling process, starting from $\hat{\mathbf{x}}_v$ (at timestep $v$) of the retrieved trajectory, to reconstruct $\mathbf{x}$. Intuitively, this RAG strategy skips the sampling process from timestep $k$ to $v$, thereby improving privacy saving, generative quality, and inference efficiency. Next, we elaborate on the implementation of RAPID's key components.

### 3.3 BUILDING TRAJECTORY KNOWLEDGE BASE

We divide the public data $\mathcal{D}^{\mathrm{pub}}$ into two parts $\mathcal{D}^{\mathrm{pub}}_{\mathrm{pre}}$ and $\mathcal{D}^{\mathrm{pub}}_{\mathrm{ref}}$ to avoid overfitting, with $\mathcal{D}^{\mathrm{pub}}_{\mathrm{pre}}$ to pre-train the diffusion model $\boldsymbol{\epsilon}_\theta$ and $\mathcal{D}^{\mathrm{pub}}_{\mathrm{ref}}$ to construct the trajectory knowledge base $\mathcal{KB}$.

For each $\mathbf{x} \in \mathcal{D}^{\mathrm{pub}}_{\mathrm{ref}}$, we construct its sampling trajectory by iteratively applying Eq. 1 to generate a sequence of latents $\{\mathbf{x}_1, \ldots, \mathbf{x}_T\}$, and store $(\mathbf{x}_k, \mathbf{x}_v)$ as a key-value pair ($k > v$) in $\mathcal{KB}$. During RAG, we may sample a random latent $\mathbf{x}_T$ at timestep $T$ and generate its early trajectory by iteratively invoking $\boldsymbol{\epsilon}_\theta$ (Eq. 3) until timestep $k$: $\{\mathbf{x}_T, \mathbf{x}_{T-1}, \ldots, \mathbf{x}_k\}$ and use $\mathbf{x}_k$ as a query to search for its nearest neighbors (in terms of $\ell_2$-norm) in $\mathcal{KB}$. For simple datasets (e.g., MNIST), due to their distributional sparsity, this straightforward approach is effective as it is possible to enforce all the trajectories to share a fixed initial latent $\mathbf{x}_T$ (Zhang et al., 2023). However, for more complex datasets (e.g., CIFAR10), their distributional density necessitates allowing different trajectories to have distinct initial latents, making this approach much less effective. More importantly, enforcing the same initial latent severely limits the model's generative quality and diversity.

Thus, instead of applying the similarity search on the latents $\{\mathbf{x}_k\}$ directly, we first extract their features (by applying a feature extractor $h$) and perform the search in their feature space. To this end, we first project $\mathbf{x}_k$ to the input space by applying one-step denoising on $\mathbf{x}_k$ using the pre-trained diffusion model $\boldsymbol{\epsilon}_\theta$, and then apply the feature extractor $h$ on the denoised $\mathbf{x}_k$ to extract its feature: $\mathbf{z}_k = h(\boldsymbol{\epsilon}_\theta(\mathbf{x}_k, k))$. For simplicity, we omit the one-step denoising in the following notations: $\mathbf{z}_k = h(\mathbf{x}_k)$.

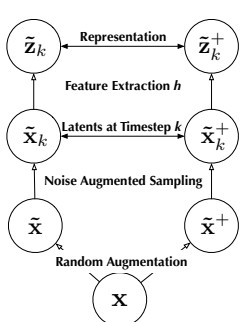

We employ contrastive learning (Chen et al., 2020b;c) to train the feature extractor $h$. Intuitively, contrastive learning learns representations by aligning the features of the same input under various augmentations (e.g., random cropping) while separating the features of different inputs. In our current implementation, we extend the SimCLR (Chen et al., 2020b) framework, as illustrated in Figure 2.

Figure 2: Contrastive learning of noise-augmented latents $\{\mathbf{x}_k\}$.

Specifically, for each input $\mathbf{x} \in \mathcal{D}^{\mathrm{pub}}_{\mathrm{ref}}$, a pair of its augmented views $(\tilde{\mathbf{x}}, \tilde{\mathbf{x}}^+)$ forms a "positive" pair,

---

**Algorithm 2:** RAPID.

---

**Input:** private data $\mathcal{D}^{\mathrm{prv}}$, pre-trained denoiser $\epsilon_\theta$, feature extractor $h$, trajectory knowledge base $\mathcal{KB}$,
  batch size $B$, timestep $k$, number of iterations $I$, gradient norm bound $C$, noise scale $\sigma$

**Output:** fine-tuned diffusion model $\epsilon_\theta$

1 **for** $i \in [I]$ **do**
2     sample a batch $\mathcal{B}$ of size $B$ from $\mathcal{D}^{\mathrm{prv}}$ via Poisson sampling;
3     **foreach** $\mathbf{x} \in \mathcal{B}$ **do**
       // find nearest neighbor
4        sample $\mathbf{x}_k$ following Eq. 3;
5        find key $\hat{\mathbf{z}}_k$ closest to $h(\mathbf{x}_k)$ in $\mathcal{KB}$;
       // skip intermediate steps
6        fetch value $\hat{\mathbf{x}}_v$ corresponding to $\hat{\mathbf{z}}_k$;
7        compute gradient $\mathbf{g}(\mathbf{x}) \leftarrow \nabla_\theta \ell_{\mathrm{DM}}(\mathbf{x}, \hat{\mathbf{x}}_v)$ following Eq. 5;
       // clip gradient
8        $\tilde{\mathbf{g}}(\mathbf{x}) \leftarrow \mathbf{g}(\mathbf{x}) / \max(1, \frac{\|\mathbf{g}(\mathbf{x})\|}{C})$;
    // apply DP noise
9     $\tilde{\mathbf{g}}(\mathcal{B}) \leftarrow \frac{1}{B} \sum_{\mathbf{x} \in \mathcal{B}} \tilde{\mathbf{g}}(\mathbf{x}) + \frac{C}{B} \mathcal{N}(\mathbf{0}, \sigma^2 \mathbf{I})$ ;
10     $\theta \leftarrow \mathrm{Adam}(\theta, \tilde{\mathbf{g}}(\mathcal{B}))$;
11 **return** $\epsilon_\theta$;

---

while a set of augmented views of other inputs $\mathcal{N}_{\mathbf{x}}^-$ forms the "negative" samples. The contrastive loss is defined by the InfoNCE loss (Oord et al., 2018), which aims to maximize the similarity of positive pairs relative to that of negative pairs:

$$\ell_{\mathrm{CL}}(\mathbf{x}) = -\log \frac{\exp(\mathrm{sim}(h(\tilde{\mathbf{x}}_k), h(\tilde{\mathbf{x}}_k^+))/\tau)}{\sum_{\tilde{\mathbf{x}}^- \in \mathcal{N}_{\mathbf{x}}^-} \exp(\mathrm{sim}(h(\tilde{\mathbf{x}}_k), h(\tilde{\mathbf{x}}_k^-))/\tau) + \exp(\mathrm{sim}(h(\tilde{\mathbf{x}}_k), h(\tilde{\mathbf{x}}_k^+))/\tau)} \quad (4)$$

where $\tilde{\mathbf{x}}_k$ denotes the sampled latent at timestep $k$ corresponding to $\tilde{\mathbf{x}}$ (similar for $\tilde{\mathbf{x}}_k^+$ and $\tilde{\mathbf{x}}_k^-$), sim is the similarity function (e.g., cosine similarity), and $\tau$ is the hyper-parameter of temperature. Algorithm 1 sketches the training of the latent feature extractor.

After training the latent feature extractor $h$, we build the trajectory knowledge base $\mathcal{KB}$. For each $\mathbf{x} \in \mathcal{D}_{\mathrm{ref}}^{\mathrm{pub}}$, we sample its trajectory as $(\mathbf{x}_1, \mathbf{x}_2, \ldots, \mathbf{x}_T)$; we consider $\mathbf{x}_k$'s feature, $\mathbf{z}_k = h(\mathbf{x}_k)$, as the key and $\mathbf{x}_v$ as the value, and store the key-value pair $(\mathbf{z}_k, \mathbf{x}_v)$ into $\mathcal{KB}$.

### 3.4 TRAINING DIFFERENTIALLY PRIVATE DIFFUSION MODEL

Leveraging the trajectory knowledge base $\mathcal{KB}$, we further train the denoiser $\epsilon_\theta$ on private data $\mathcal{D}^{\mathrm{prv}}$. Notably, we focus on training $\epsilon_\theta$ from timestep $v$ to 0, leading to the advantages of fully utilizing limited private data and reducing overall privacy costs.

As outlined in Algorithm 2, at each iteration, we sample batch $\mathcal{B}$ from $\mathcal{D}^{\mathrm{prv}}$ using Poisson sampling for privacy amplification (Mironov et al., 2019). For each input $\mathbf{x} \in \mathcal{B}$, we (*i*) sample its early trajectory $\mathbf{x}_k$ up to timestep $k$, (*ii*) use its feature $\mathbf{z}_k = h(\mathbf{x}_k)$ as a query to find $\mathbf{z}_k$'s nearest neighbor $\hat{\mathbf{z}}_k$ in $\mathcal{KB}$, (*iii*) reuse the value $\hat{\mathbf{x}}_v$ corresponding to $\hat{\mathbf{z}}_k$ in $\mathcal{KB}$ as the starting point at timestep $v$, and (*iv*) train $\epsilon_\theta$ to reconstruct $\mathbf{x}$. In other words, $\epsilon_\theta$ is fine-tuned to predict the random noise $(\mathbf{x} - \hat{\mathbf{x}}_v)$ at timestep $v$. To make the training differentially private, we extend DP-SGD (Abadi et al., 2016) during updating $\epsilon_\theta$. For each input $\mathbf{x}$ in a batch $\mathcal{B}$, we compute its diffusion loss as:

$$\ell_{\mathrm{DM}}(\mathbf{x}, \hat{\mathbf{x}}_v) = \mathbb{E}_{v' \sim \mathcal{U}(1,v)} \Big\| \frac{\hat{\mathbf{x}}_v - \sqrt{\bar{\alpha}_v} \mathbf{x}}{\sqrt{1 - \bar{\alpha}_v}} - \epsilon_\theta \Big( \sqrt{\bar{\alpha}_{v'}} \mathbf{x} + \frac{\sqrt{1 - \bar{\alpha}_{v'}}}{\sqrt{1 - \bar{\alpha}_v}} (\hat{\mathbf{x}}_v - \sqrt{\bar{\alpha}_v} \mathbf{x}), v' \Big) \Big\| \quad (5)$$

where the random noise $(\mathbf{x} - \hat{\mathbf{x}}_v)$ is scaled with a randomly sampled timestep $v'$. We compute the gradient $\mathbf{g}(\mathbf{x}) = \nabla_\theta \ell_{\mathrm{DM}}(\mathbf{x}, \hat{\mathbf{x}}_v)$. To bound $\mathbf{g}(\mathbf{x})$'s influence on $\epsilon_\theta$, we clip $\mathbf{g}(\mathbf{x})$ using its $\ell_2$ norm. We then sanitized the per-batch gradient as $\tilde{\mathbf{g}}(\mathcal{B})$ by applying random Gaussian noise before updating $\epsilon_\theta$ using the Adam optimizer (Kingma & Ba, 2014).

We prove the privacy guarantee of Algorithm 2 under Rényi differential privacy (RDP) (Mironov, 2017), which can be converted to $(\epsilon, \delta)$-DP. The following theorem formulates the guarantee provided by RAPID (proof deferred to §A).

**Theorem 1.** *Using the sanitized per-batch gradient $\tilde{\mathbf{g}}(\mathcal{B})$ to update $\epsilon_\theta$ satisfies $(\alpha, \frac{2\alpha}{\sigma^2})$-RDP.*

The overall privacy cost of RAPID is computed via RDP composition (Mironov, 2017), which can be further improved using more advanced privacy accounting (Gopi et al., 2021).

The inference of RAPID runs as follows. By sampling random Gaussian noise $\mathbf{x}_T$ at timestep $T$, we generate its early trajectory $\mathbf{x}_k$ up to timestep $k$; using the feature $\mathbf{z}_k$ of $\mathbf{x}_k$ as the query, we search for $\mathbf{z}_k$'s nearest neighbor $\hat{\mathbf{z}}_k$ in $\mathcal{KB}$; we then use the corresponding value $\hat{\mathbf{x}}_v$ as the starting point at timestep $v$ to resume the sampling. Compared with prior work (Lyu et al., 2023; Dockhorn et al., 2023), this RAG-based inference also significantly improves inference efficiency.

## 4 EVALUATION

### 4.1 EXPERIMENTAL SETTING

**Datasets.** We focus on the image synthesis task. In each task, we use the public dataset $\mathcal{D}^{\mathrm{pub}}$ to pre-train the diffusion model and build the trajectory knowledge base for the retrieval-augmented generation, and use the private dataset $\mathcal{D}^{\mathrm{pub}}$ to further fine-tune/train the diffusion model in a differentially private manner. Specifically, we consider the following 4 settings: i) EMNIST (Cohen et al., 2017) (public) and MNIST (Deng, 2012) (private), ii) ImageNet32 Deng et al. (2009) (public) and CIFAR10 (Krizhevsky et al., 2009) (private), iii) FFHQ32 (Karras et al., 2019) (public) and CelebA32 (Liu et al., 2015) (private), and iv) FFHQ64 (Karras et al., 2019) (public) and CelebA64 (Liu et al., 2015) (private). More details of these datasets are deferred to Table 5.

**Diffusion models.** We primarily use the latent diffusion model (Rombach et al., 2022) as the underlying diffusion model and DDIM (Song et al., 2020) as the default sampler.

**Baselines.** We mainly consider two state-of-the-art DPDM methods as baselines: differential private diffusion model (DPDM) (Dockhorn et al., 2023) and differential private latent diffusion model (DP-LDM) Lyu et al. (2023).

**Metrics.** Following prior work (Dockhorn et al., 2023; Lyu et al., 2023), we use the Frechet Inception Distance (FID) score to measure the generative quality of different methods. In addition, we adopt the coverage metric (Naeem et al., 2020) to measure the generative diversity. Intuitively, given a reference real dataset $\mathcal{D}^{\mathrm{real}}$, the coverage is measured by the proportion of samples from $\mathcal{D}^{\mathrm{real}}$ that have at least one sample from the synthesized data $\mathcal{D}^{\mathrm{syn}}$ in their neighborhood (with neighborhood size fixed as 5 (Lebensold et al., 2024)). Formally,

$$\mathrm{Coverage} = \frac{1}{|\mathcal{D}^{\mathrm{real}}|} \sum_{\mathbf{x} \in \mathcal{D}^{\mathrm{real}}} \mathbf{1}_{\exists \mathbf{x}' \in \mathcal{D}^{\mathrm{syn}} \wedge \mathbf{x}' \in \mathcal{N}_{\mathbf{x}}} \tag{6}$$

where $\mathbf{1}$ is the indicator function and $\mathcal{N}_{\mathbf{x}}$ denotes $\mathbf{x}$'s neighborhood.

**Privacy.** We use OPACUS Yousefpour et al. (2021), a DP-SGD library, for DP training and privacy accounting. Following prior work (Dockhorn et al., 2023), we fix the setting of $\delta$ as $10^{-5}$ for the CIFAR10 and MNIST datasets and $10^{-6}$ for CelebA dataset so that $\delta$ is smaller than the reciprocal of the number of training samples. Similar to existing work, we also do not account for the (small) privacy cost of hyper-parameter tuning.

To simulate settings with modest compute resources, all the experiments are performed on a workstation running one Nvidia RTX 6000 GPU.

### 4.2 MAIN RESULTS

We empirically evaluate RAPID and baselines. To make a fair comparison, we fix the default batch size as 64 for RAPID and DP-LDM; we do not modify the batch size (i.e., 8,192) for DPDM because the impact of batch size on its performance is so significant that it stops generating any recognizable images with smaller batch sizes. By default, we fix the sampling timesteps as 100 across all the methods. For MNIST, we train the diffusion model under three privacy settings $\epsilon = \{0.2, 1, 10\}$, corresponding to the low, medium, and high privacy budgets; for the other datasets, we vary the privacy budget as $\epsilon = \{1, 10\}$.

**Class-conditional generation.** We evaluate the quality of class-conditional generation by different methods, in which, besides the input image $\mathbf{x}$, a guidance signal $y$ (e.g., $\mathbf{x}$'s class label) is also pro-

| Setting | Privacy ($\epsilon$) | DPDM | DP-LDM | RAPID |
|---------|------------|------|--------|-------|
| EMNIST→MNIST | 0.2 | 125.7 | 50.8 | 24.0 |
|  | 1 | 50.5 | 34.9 | 18.5 |
|  | 10 | 12.9 | 27.2 | 14.1 |
| ImageNet32→CIFAR10 | 1 | \ | 79.1 | 63.2 |
|  | 10 | 109.9 | 33.3 | 25.4 |

Table 1. FID scores of class-conditional generation by different methods.

vided for training (and inference). We consider the EMNIST→MNIST and ImageNet32→CIFAR10 settings. Table 1 compares the generative quality of different methods. Observe that, under the same privacy budget, RAPID considerably outperforms the baselines across most cases. For instance, under the ImageNet32→CIFAR10 setting, with $\epsilon = 1$, RAPID attains an FID score of 63.2, while DPDM fails to produce any sensible outputs, highlighting the effectiveness of RAG in facilitating the DP training of diffusion models.

| Setting | Privacy ($\epsilon$) | DPDM | DP-LDM | RAPID |
|---------|------------|------|--------|-------|
| EMNIST→MNIST (CNN) | 0.2 | 85.77% | 11.35% | 96.43% |
|  | 1 | 95.18% | 74.62% | 98.11% |
|  | 10 | 98.06% | 95.54% | 99.04% |
| ImageNet32→CIFAR10 (ResNet) | 1 | \ | 50.39% | 63.61% |
|  | 10 | 30.41% | 66.02% | 67.37% |

Table 2. Downstream accuracy of classifiers trained on synthesized data.

We further evaluate the utility of the data synthesized by different methods to train downstream classifiers. For a fair comparison, we use the same classifier architecture to measure the downstream accuracy. For the synthesized MNIST data, we train a Convolutional Neural Network (CNN) (Krizhevsky et al., 2012) and test its performance on the MNIST testing set. For the synthesized CIFAR10 data, we train a ResNet-9 (He et al., 2016) and evaluate its accuracy on the CIFAR10 testing set, with results summarized in Table 2. Observe that the classifier trained on RAPID's synthesized data largely outperforms the other methods. For instance, under the ImageNet32→CIFAR10 setting with $\epsilon = 1$, RAPID attains 63.6% downstream accuracy with 10% higher than the baselines, indicating the high utility of the data synthesized by RAPID.

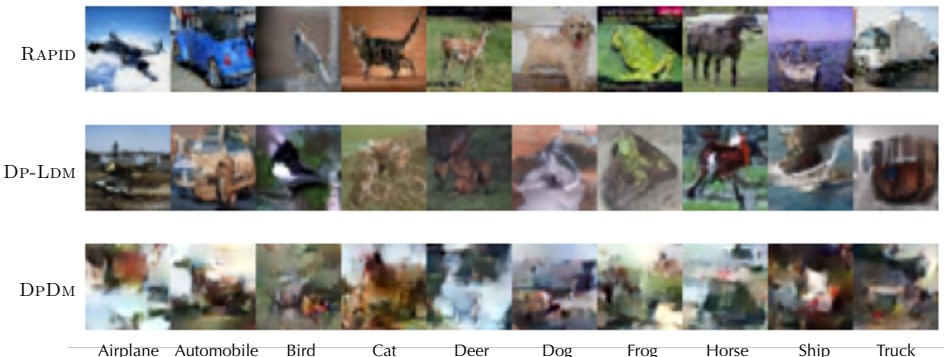

Figure 3: Random samples synthesized by RAPID and baselines trained under the ImageNet32→CIFAR10 setting with $\epsilon = 10$.

We also qualitatively compare the class-conditional samples generated by DPDM, DP-LDM, and RAPID trained under the ImageNet32→CIFAR10 setting (with $\epsilon = 10$), as shown in Figure 3. It is observed that across different classes, RAPID tends to produce samples of higher visual quality, compared with the baselines.

**Unconditional generation.** We further evaluate the unconditional generation by different methods under the FFHQ32→CelebA32 and FFHQ64→CelebA64 settings. Table 3 summarizes the FID and coverage scores of different methods. Observe that RAPID outperforms the baselines by large margins in terms of generative quality (measured by the FID score). For instance, in the case of FFHQ32→CelebA32 under $\epsilon = 10$, RAPID achieves an FID score of 37.3, which is 19.1% and 51.8% lower than DP-LDM and DPDM, respectively, highlighting its superior generative quality. Meanwhile, across all the cases, RAPID attains the highest (or the second highest) generative di-

| Setting | Privacy ($\epsilon$) | DPDM | DP-LDM | RAPID |
|---|---|---|---|---|
| FFHQ32→CelebA32 | 1 | 135.9 — 0.087 | 65.3 — 0.74 | 52.8 — 0.96 |
| | 10 | 29.8 — 0.55 | 38.0 — 0.98 | 28.0 — 0.98 |
| FFHQ64→CelebA64 | 1 | \ | 72.2 — 0.59 | 60.5 — 0.90 |
| | 10 | 80.8 — 0.094 | 45.2 — 0.94 | 37.3 — 0.93 |

Table 3. FID (left) and coverage (right) scores of unconditional generation by different methods.

versity (measured by the coverage score). Overall, RAPID strikes the optimal balance between generative quality and diversity among all three methods.

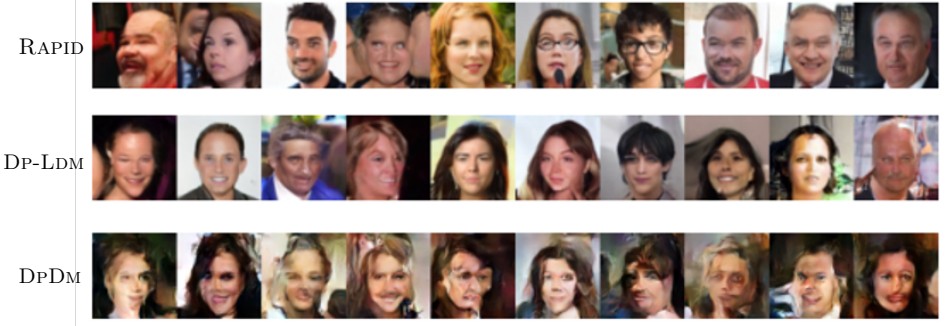

Figure 4: Random samples synthesized by RAPID and baselines trained under the setting of FFHQ64→CelebA64 (with $\epsilon = 10$).

We also qualitatively compare the unconditional samples generated by different methods trained on the CelebA64 dataset (with $\epsilon = 10$). Figure 3 shows random samples synthesized by DPDM, DP-LDM, and RAPID (more samples in §C). Observe that in general RAPID tends to produce unconditional samples of higher visual quality, compared with the baselines.

## 4.3 ABLATION STUDIES

Next, we conduct ablation studies to understand the impact of various key factors, such as batch size and knowledge base size, on RAPID's performance. We use the FFHQ32→CelebA32 (with $\epsilon = 10$) as the default setting.

**Retrieval accuracy.** Recall RAPID relies on retrieving the most similar trajectory from the knowledge base. A crucial question is thus whether RAPID indeed retrieves semantically relevant neighbors. To answer this question, under the class-conditional generation, we evaluate the accuracy of RAPID in retrieving the neighbors from the class corresponding to the given label (e.g., "automobile"). We calculate the top-$k$ ($k = 1, 5$) accuracy based on the true labels of the retrieved neighbors. Under the setting of ImageNet32→CIFAR10, RAPID attains 81.2% top-1 accuracy and 94.4% top-5 accuracy, respectively, indicating its effectiveness.

**Batch size.** In contrast to existing methods (e.g., DPDM and DP-LDM) that typically require excessively large batch sizes (e.g., 8,192 samples per batch), by fully utilizing the public data using its RAG design, RAPID can generate high-quality samples under small batch sizes. Here, we evaluate the impact of batch-size setting on the performance of RAPID and DP-LDM, with results illustrated in Figure 5. It can be noticed that RAPID attains an FID score of 29.5 under a batch size of 16 while its score steadily improves to 26.67 as the batch size increases from 16 to 256, highlighting its superior performance under small batch sizes.

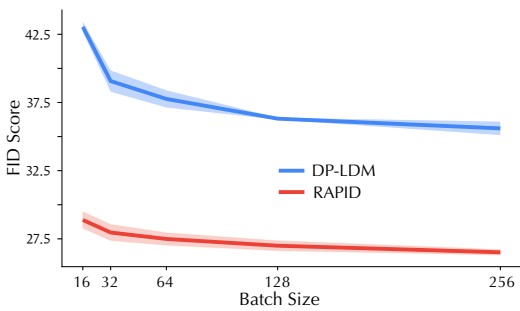

Figure 5: Impact of batch size.

**Sampler.** By default, we use DDIM (Song et al., 2020) as the underlying sampler. Here, we evaluate the influence of the sampler on RAPID. Specifically, we consider the state-of-the-art PNDM sampler (Liu et al., 2022) and evaluate the performance of different methods in unconditional generation

| Setting | Privacy ($\epsilon$) | DPDM | DP-LDM | RAPID |
|---|---|---|---|---|
| FFHQ32→CelebA32 | 1 | 153.1 | 72.2 | 56.6 |
| | 10 | 33.0 | 42.6 | 30.3 |
| FFHQ64→CelebA64 | 1 | \ | 78.7 | 68.9 |
| | 10 | 86.2 | 50.3 | 41.1 |

Table 4. FID scores of unconditional generation with the PNDM sampler.

tasks. By comparing Table 3 and Table 4, it is observed that RAPID achieves similar FID scores in both cases, indicating its insensitivity to the underlying sampler.

**Knowledge base size.** One key component of RAPID is its trajectory knowledge base that supports RAG. We now evaluate the impact of the knowledge base size. As shown in Figure 6, we evaluate how RAPID's performance varies as the knowledge-base size grows from 100 to 50,000. As expected, both RAPID's FID and coverage scores improve greatly with the knowledge-base size. Meanwhile, even under a small knowledge base (e.g., of size 100), RAPID attains satisfactory performance (e.g., with an FID score of about 40 and a coverage of about 0.9).

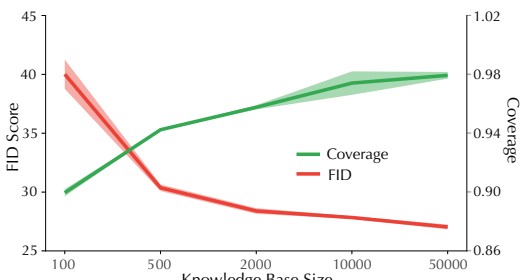

Figure 6: Impact of knowledge-base size.

**Fraction of privatized steps.** Recall that, over the sample trajectory, RAPID samples the initial $(T - k)$ steps, skips the intermediate $(k - v)$ steps, and privatizes the later $v$ steps. By default, we set $k/T = 0.8$ and $v/T = 0.2$. We now evaluate the influence of the fraction of privatized steps $v/T$ on RAPID. Figure 7 shows how RAPID's FID score varies as $v/T$ increases from 0.3 to 0.7 (with $k/T$ fixed as 0.8) under

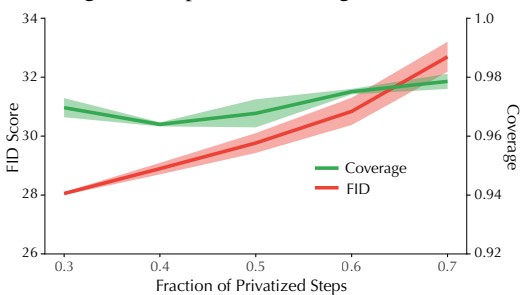

Figure 7: Impact of the fraction of privatized steps.

the FFHQ32→CelebA32 setting (with $\epsilon = 10$). Notably, as more steps are privatized, RAPID's FID score deteriorates while its coverage score improves marginally, suggesting an interesting trade-off between the generative quality and diversity. Intuitively, a larger fraction of $v/T$ indicates less reliance on the retrieved trajectory, encouraging more diverse generations but negatively impacting the generative quality.

## 5 CONCLUSION AND FUTURE WORK

The work represents a pilot study of integrating retrieval-augmented generation (RAG) into the private training of generative models. We present RAPID, a novel approach for training differentially private (DP) diffusion models. Through extensive evaluation using benchmark datasets and models, we demonstrate that RAPID largely outperforms state-of-the-art methods in terms of generative quality, memory footprint, and inference efficiency. The findings suggest that integrating RAG with DP training represents a promising direction for designing privacy-preserving generative models.

This work also opens up several avenues for future research. (*i*) Like other approaches for training DP diffusion models and the broader pre-training/fine-tuning paradigm, RAPID relies on access to a diverse public dataset that captures a range of patterns and shares similar high-level layouts with the private data. It is worth exploring scenarios with highly dissimilar public/private data (Liu et al., 2021a;b; Fuentes et al., 2024). (*ii*) In its current implementation, RAPID retrieves only the top-1 nearest trajectory in RAG. Exploring ways to effectively aggregate multiple neighboring trajectories could improve generative quality and diversity. (*iii*) While RAPID's privacy accounting focuses on privatizing the fine-tuning stage, it is worth accounting for random noise introduced by the diffusion process to further improve its privacy guarantee (Wang et al., 2024). (*iv*) Although this work primarily focuses on image synthesis tasks, given the increasingly widespread use of diffusion models, extending RAPID to other tasks (e.g., text-to-video generation) presents an intriguing opportunity.

ACKNOWLEDGMENT

This work is partially supported by the National Science Foundation under Grant No. 2405136, 2406572, and 2212323.

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

## A  PROOFS

*Definition* 1. (Rényi differential privacy (Mironov, 2017)) A randomized mechanism $\mathcal{M} : \mathcal{D} \to \mathcal{R}$ over domain $\mathcal{D}$ and range $\mathcal{R}$ satisfies $(\alpha, \epsilon)$-RDP if for any two adjacent $d, d' \in D$: $\mathcal{D}_\alpha(\mathcal{M}(d)|\mathcal{M}(d')) \leq \epsilon$, where $\mathcal{D}_\alpha$ denotes the Rényi divergence of order $\alpha$.

RDP can be converted to DP. If an mechanism satisfies $(\alpha, \epsilon)$-RDP, it also satisfies $(\epsilon + \frac{\log 1/\delta}{\alpha-1}, \delta)$-DP (Mironov, 2017).

Notably, Gaussian mechanism can provide RDP. Specifically, for function $f$ with sensitivity $\Delta f = \max_{d,d'} \|f(d) - f(d')\|_2$, releasing $f(d) + \mathcal{N}(0, \sigma^2)$ satisfies $(\alpha, \frac{\alpha \Delta f}{2\sigma^2})$-RDP (Mironov et al., 2019).

Now, we prove Theorem 1. Recall that Algorithm 2 computes the per-sample gradient $\mathbf{g}(\mathbf{x})$ and clips it to bound its influence $\tilde{\mathbf{g}}(\mathbf{x}) \leftarrow \mathbf{g}(\mathbf{x})/\max(1, \frac{\|\mathbf{g}(\mathbf{x})\|}{C})$. It then computes the per-batch gradient $\mathbf{g}(\mathcal{B}) \leftarrow \frac{1}{B} \sum_{\mathbf{x} \in \mathcal{B}} \tilde{\mathbf{g}}(\mathbf{x})$ and applies Gaussian noise: $\tilde{\mathbf{g}}(\mathcal{B}) \leftarrow \mathbf{g}(\mathcal{B}) + \frac{C}{B} \mathcal{N}(0, \sigma^2 \mathbf{I})$.

*Proof.* Consider two adjacent mini-batches $\mathcal{B}$ and $\mathcal{B}'$ that differ by one sample $\mathbf{x}^-/\mathbf{x}^+$: $\mathcal{B}' = \mathcal{B} \setminus \{\mathbf{x}^-\} \cup \{\mathbf{x}^+\}$. We bound the difference of their batch-level gradients as follows:

$$
\begin{aligned}
\|\mathbf{g}(\mathcal{B}) - \mathbf{g}(\mathcal{B}')\|_2 &= \|\frac{1}{B} \sum_{\mathbf{x} \in \mathcal{B}} \tilde{\mathbf{g}}(\mathbf{x}) - \frac{1}{B} \sum_{\mathbf{x} \in \mathcal{B}'} \tilde{\mathbf{g}}(\mathbf{x})\|_2 \\
&= \|\frac{1}{B} \tilde{g}(\mathbf{x}^-) - \frac{1}{B} \tilde{g}(\mathbf{x}^+)\|_2 \\
&= \frac{1}{B} \sqrt{\|\tilde{g}(\mathbf{x}^-)\|_2^2 + \|\tilde{g}(\mathbf{x}^+)\|_2^2 - 2\tilde{g}^T(\mathbf{x}^-)\tilde{g}(\mathbf{x}^+)} \\
&\leq \frac{1}{B} \sqrt{C^2 + C^2 + 2C^2} \\
&= \frac{2C}{B}
\end{aligned}
\tag{7}
$$

where we use the fact that $\tilde{g}(\mathbf{x}^-)$ and $\tilde{g}(\mathbf{x}^+)$ are bounded by $C$ and the Cauchy-Schwarz inequality.

Thus, the sensitivity of $g(\mathcal{B})$ is $\frac{2C}{B}$. Following the RDP Gaussian mechanism, releasing the sanitized batch-level gradient $\tilde{g}(\mathcal{B})$ provides $(\alpha, \frac{2\alpha}{\sigma^2})$-RDP, corresponding to $(\frac{2\alpha}{\sigma^2} + \frac{\log 1/\delta}{\alpha-1}, \delta)$-DP. □

## B  EXPERIMENTAL SETTING

Table 5 summarizes the setting of public and private datasets in our experiments.

| Public dataset $\mathcal{D}^{\text{pub}}$ | | Private dataset $\mathcal{D}^{\text{prv}}$ |
| --- | --- | --- |
| Pre-training ($\mathcal{D}^{\text{pub}}_{\text{pre}}$) | Trajectory knowledge base $\mathcal{D}^{\text{pub}}_{\text{ref}}$ | |
| EMNIST (50K) | EMNIST (10K) | MNIST |
| ImageNet32 (1.2M) | ImageNet32 (70K) (Darlow et al., 2018) | CIFAR10 |
| FFHQ32 (60K) | FFHQ32 (10K) | CelebA32 |
| FFHQ64 (60K) | FFHQ64 (10K) | CelebA64 |

Table 5. Setting of public/private datasets in experiments.

Table 6 lists the default parameter setting for training the autoencoder in the latent diffusion model under different settings, while Table 7 summarizes the default parameter setting for training the diffusion model.

## C  ADDITIONAL RESULTS

### C.1  QUALITATIVE COMPARISON

Figure 8 illustrates random samples synthesized by RAPID and baselines under the setting of FFHQ32→CelebA32 (with $\epsilon = 10$).

|  | EMNIST→MNIST | ImageNet→CIFAR10 | FFHQ32→CelebA32 | FFHQ64→CelebA64 |
|---|---|---|---|---|
| Input size | 32×32×3 | 32×32×3 | 32×32×3 | 64×64×3 |
| $z$-shape | $4 \times 4 \times 3$ | $16 \times 16 \times 3$ | $16 \times 16 \times 3$ | $64 \times 64 \times 3$ |
| Channels | 128 | 128 | 128 | 192 |
| Channel multiplier | $[1, 2, 3, 5]$ | $[1, 2]$ | $[1, 2]$ | $[1, 2]$ |
| Attention resolutions | $[32, 16, 8]$ | $[16, 8]$ | $[16, 8]$ | $[16, 8]$ |
| # ResBlocks | 2 | 2 | 2 | 2 |
| Batch size | 64 | 64 | 64 | 64 |

Table 6. Hyper-parameters for training autoencoders under different settings.

|  | EMNIST→MNIST | ImageNet→CIFAR10 | FFHQ32→CelebA32 | FFHQ64→CelebA64 |
|---|---|---|---|---|
| Input size | 32×32×3 | 32×32×3 | 32×32×3 | 64×64×3 |
| $z$-shape | $4 \times 4 \times 3$ | $16 \times 16 \times 3$ | $16 \times 16 \times 3$ | $32 \times 32 \times 3$ |
| # Channels | 64 | 128 | 192 | 192 |
| Channel multiplier | $[1, 2]$ | $[1, 2, 2, 4]$ | $[1, 2, 4]$ | $[1, 2, 4]$ |
| Attention resolutions | $[1, 2]$ | $[1, 2, 4]$ | $[1, 2, 4]$ | $[1, 2, 4]$ |
| # ResBlocks | 1 | 2 | 2 | 2 |
| # Heads | 2 | 8 | 8 | 8 |
| Batch size | 64 | 64 | 64 | 32 |
| Spatial transformer | True | True | False | False |
| Cond_stage_key | class_label | class_label | class_label | class_label |
| Conditioning_key | crossattn | crossattn | crossattn | crossattn |
| # Classes | 26 | 1000 | 1000 | 1000 |
| Embedding dimensions | 5 | 512 | 512 | 512 |
| Transformer depth | 1 | 2 | 2 | 2 |

Table 7. Hyper-parameters for diffusion models under different settings.

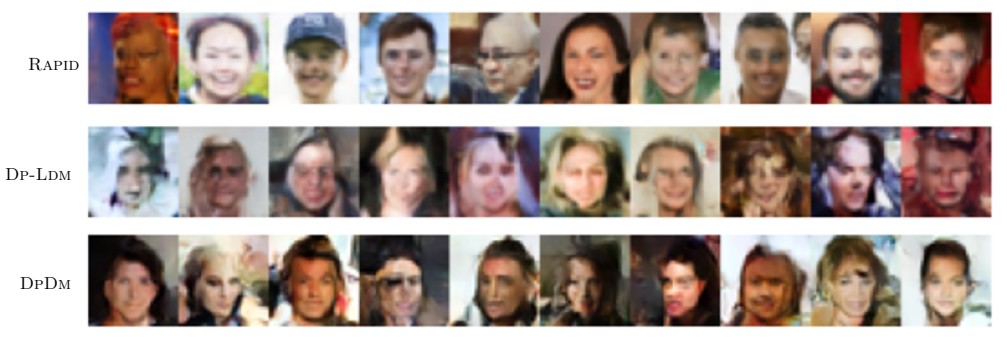

Figure 8: Random samples synthesized by RAPID and baselines trained under the FFHQ32→CelebA32 setting (with $\epsilon = 10$).

## C.2 DISSIMILAR PUBLIC/PRIVATE DATA

To evaluate RAPID's robustness to the distributional shift between public and private data, we conduct additional experiments to evaluate RAPID's performance when using dissimilar public/private datasets. Specifically, we use ImageNet32 as the public dataset with added Gaussian noise $\mathcal{N}(0, 0.1)$ to degrade its quality. For the private dataset, we use VOC2005 (Everingham, 2005) (resized to $32\times32$), a dataset used for object detection challenges in 2005, which significantly differs from ImageNet32 and contains only about 1K images. We apply RAPID in this challenging setting, with results shown in Table 8. Notably, RAPID outperforms baselines (e.g., DP-LDM) in terms of FID scores across varying $\epsilon$, indicating its robustness to dissimilar public/private datasets.

| Privacy ($\epsilon$) | DP-LDM | DPSDA | PRIVIMAGE | RAPID |
|---|---|---|---|---|
| 1 | 164.85 | 142.20 | 139.07 | 93.17 |
| 10 | 147.86 | 130.42 | 123.89 | 82.56 |

Table 8. Performance of RAPID and baselines (measured by FID scores) in the ImageNet32→VOC2005 case.

Moreover, we compare RAPID's performance (without DP) to direct training on the VOC2005 dataset. RAPID improves the FID score from 77.83 to 54.60, highlighting its ability to effectively leverage the public data even when it differs substantially from the private data.

## C.3 ADDITIONAL BASELINES

We further compare RAPID with more recent work on DP diffusion models. DPSDA (Lin et al., 2024) synthesizes a dataset similar to the private data by iteratively querying commercial image generation APIs (e.g., DALL-E 2) in a DP manner. For fair comparison with RAPID, instead of using commercial APIs trained on vast datasets (hundreds of millions of images), following the setting of Li et al. (2024) that replicates DPSDA's results, we use ImageNet32 for pre-training the public model (also as the query API for DPSDA) and CIFAR10 as the private dataset.

| Privacy ($\epsilon$) | Model Size = 90M | | Model Size = 337M | |
|---|---|---|---|---|
| | DPSDA | RAPID | DPSDA | RAPID |
| 1 | 113.6 | 63.2 | 89.1 | 66.5 |
| 10 | 60.9 | 25.4 | 43.8 | 29.0 |

Table 9. Performance of DPSDA and RAPID (measured by FID scores) in the ImageNet32→CIFAR10 case.

Note RAPID and DPSDA represent two distinct approaches to training DP diffusion models, with the pre-trained model size affecting their performance differently.

For DPSDA, which uses DP evolution rather than DP training to synthesize data, larger pre-trained models tend to lead to better performance. This is demonstrated in DPSDA's ablation study (Lin et al., 2024), where increasing the model size from 100M to 270M parameters improves results by enhancing the quality of selected data. In contrast, methods involving DP training (such as DPDM, DP-LDM, PRIVIMAGE, and RAPID) may not benefit from heavily over-parameterized models, as shown in (Dockhorn et al., 2023). This is because the $\ell_2$-norm noise added in DP-SGD typically grows linearly with the number of parameters.

To empirically evaluate how model complexity affects different approaches, we conduct experiments varying the size of the pre-trained model from 90M to 337M parameters (by increasing the latent diffusion model's architecture from 128 to 192 channels and expanding its residual blocks from 2 to 4). Table 9 compares the performance (measured by FID scores) of DPSDA and RAPID across different pre-trained model sizes. As model complexity increases, DPSDA achieves better FID scores, while RAPID shows only marginal performance degradation. Notably, when using the same public dataset and pre-trained model, RAPID consistently outperforms DPSDA, suggesting that it is more effective at leveraging public data under DP constraints.

PRIVIMAGE (Li et al., 2024) uses the fine-tuning approach, querying the private data distribution to select semantically similar public samples for pretraining, followed by DP-SGD fine-tuning on the private data. The table below compares RAPID and PRIVIMAGE's performance across different $\epsilon$ values on CIFAR10 and CelebA64.

Notably, RAPID outperforms PRIVIMAGE in most scenarios, with one exception: CIFAR10 under $\epsilon = 1$. This likely occurs because PRIVIMAGE selects public data similar to the private data for pre-training. With clearly structured private data (for instance, CIFAR10 contains 10 distinct classes),

| Privacy ($\epsilon$) | CIFAR10 | | CelebA64 | |
|---|---|---|---|---|
| | PRIVIMAGE | RAPID | PRIVIMAGE | RAPID |
| 1 | 29.8 | 63.2 | 71.4 | 60.5 |
| 10 | 27.6 | 25.4 | 49.3 | 37.3 |

Table 10. Performance comparison of PRIVIMAGE and RAPID (measured by FID scores).

using a targeted subset rather than all the public data tends to improve DP fine-tuning, especially under strict privacy budgets. However, this advantage may diminish with less structured private data (e.g., CelebA64). We consider leveraging the PRIVIMAGE's selective data approach to enhance RAPID as our ongoing research.

## C.4 IMPACT OF RETRIEVAL-AUGMENTED TRAINING

RAPID can integrate with existing methods for training DP diffusion models since it is agnostic to model training, though its neighbor retrieval operates on latents, making it compatible only with latent diffusion models. To measure the impact of RAPID, we use a latent diffusion model as the backbone model for both DP-LDM and DPDM, evaluating their performance with and without RAPID. Table 11 shows results on MNIST and CIFAR10 at $\epsilon = 10$. The substantial FID score improvement demonstrates the effectiveness of retrieval-augmented training.

| Dataset | DPDM | | DP-LDM | |
|---|---|---|---|---|
| | w/o | w/ | w/o | w/ |
| MNIST | 42.9 | 25.4 | 27.2 | 14.1 |
| CIFAR10 | 82.2 | 54.1 | 33.3 | 25.4 |

Table 11. Impact of retrieval-augmented training on existing methods ($\epsilon = 10$).

## C.5 KNOWLEDGE BASE GENERATION

While prior work on retrieval augmented generation (e.g., REDI Zhang et al. (2023)) requires all the trajectories to share the same latent, building the knowledge base needs to iteratively sample tens of thousands of trajectories from a pre-trained diffusion model (e.g., Stable Diffusion), which is highly expensive. For instance, on a workstation running one Nvidia RTX 6000 GPU, REDI requires over 8 hours to build a 10K-sample knowledge base.

| Knowledge Base Size | 10K | 20K | 30K | 40K | 50K | 60K | 70K |
|---|---|---|---|---|---|---|---|
| RAPID | 2.10s | 4.17s | 6.12s | 8.23s | 10.33s | 12.19s | 14.48s |

Table 12. Runtime of RAPID for knowledge base construction.

In comparison, RAPID eliminates this constraint, which allows it to directly compute the trajectory for each sample in the public dataset in a forward pass. Table 12 shows RAPID's runtime efficiency for various knowledge base sizes, achieving orders of magnitude faster performance than prior work.

## C.6 PERFORMANCE WITH VARYING $\epsilon$

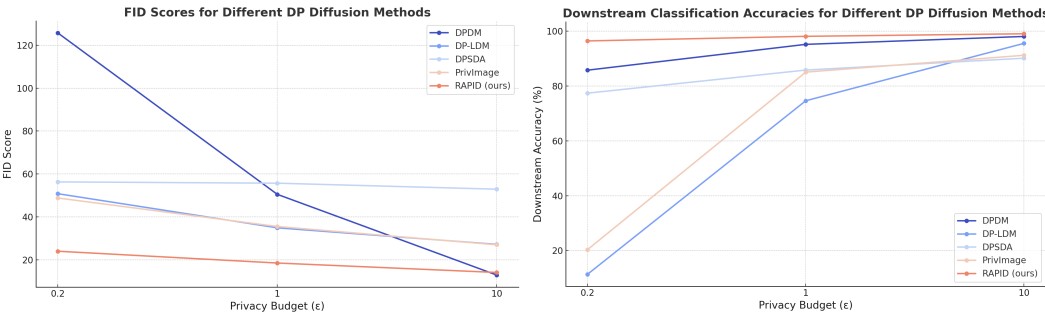

Figure 9: Performance of RAPID and baselines with varying $\epsilon$ on EMNIST→MNIST: (a) FID scores and (b) downstream classification accuracy

Figure 9 compares the performance (measured by FID scores and downstream classification accuracy) of RAPID and baselines (DPDM, DP-LDM, DPSDA, and PRIVIMAGE) in the case of EMNIST→MNIST under varying $\epsilon$ settings. Observe that, under the same privacy budget, RAPID considerably outperforms the baselines across most cases.

# D  DISCUSSION

## D.1  COMPARISON OF RAPID AND REDI

REDI Zhang et al. (2023) also employs some strategies similar to RAPID such as constructing trajectory knowledge bases at early stages to bypass intermediate steps in the generation process. However, the two methods differ in several fundamental aspects.

First, REDI employs RAG in the inference stage to improve generative efficiency, while RAPID integrates RAD into the DP training of diffusion models. Second, unlike REDI that builds its knowledge base by iteratively sampling tens of thousands of diffusion trajectories from a pre-trained latent diffusion model (e.g., Stable Diffusion), RAPID constructs the knowledge base by directly computing the diffusion trajectories via adding a scaled version of the initial latent to each sample in the public dataset, which greatly reduces the computational cost. Last, all the trajectories in REDI share the same initial latent. In contrast, the initial latents in RAPID are randomly sampled, significantly improving the diversity of generated samples.

## D.2  IMPACT OF PUBLIC/PRIVATE DATA SIMILARITY

Like other DP diffusion model approaches (e.g., DPDM, DP-LDM, PRIVIMAGE) and the broader pre-training/fine-tuning paradigm, RAPID assumes access to a diverse public dataset that captures a range of patterns. However, RAPID is more flexible: the public and private datasets need not closely match in distribution, as long as the public dataset contains similar high-level layouts. Here, we explore the possible explanations.

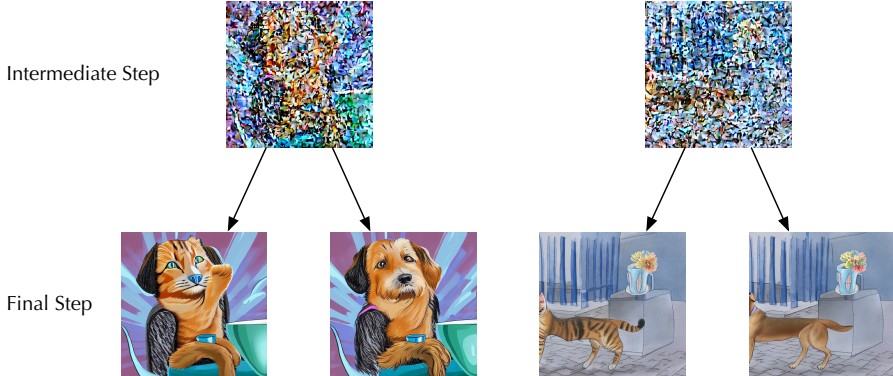

Figure 10: Disentanglement effects of diffusion models.

Existing studies (Meng et al., 2021; Zhang et al., 2023) establish that in diffusion models, early stages determine image layouts that can be shared across many generation trajectories, while later steps define specific details. Wu et al. (2022) further discover diffusion models' disentanglement capability, allowing generation of images with different styles and attributes from the same intermediate sampling stage, as shown in Figure 10. This disentanglement property enables RAPID to maintain robust performance even when public and private dataset distributions differ significantly, provided their high-level layouts remain similar.

