# OpenReview forum: "RAPID: Retrieval Augmented Training of Differentially Private Diffusion Models"
_ICLR.cc/2025/Conference — ICLR 2025 Poster_

### Official Review · Reviewer_iaJi · 2024-11-03

**Soundness:** 3
**Presentation:** 3
**Contribution:** 3
**Rating:** 6
**Confidence:** 3

**Summary:**

This paper introduces a novel method called RAPID that integrates retrieval-augmented generation (RAG) into the training of differentially private diffusion models (DPDMs). This approach leverages public data to create a knowledge base of sample trajectories, which are used as surrogates during the training on private data. By focusing differential privacy constraints on the later sampling steps and reusing similar public trajectories for the initial steps, RAPID significantly improves the trade-off between privacy and utility. The method improves existing DPDMs by reducing utility loss, memory footprint, and inference costs. Evaluations on benchmark datasets demonstrate that RAPID outperforms state-of-the-art methods in generative quality, memory efficiency, and inference speed. This work suggests that integrating RAG with differentially private training represents a promising direction for developing privacy-preserving generative models.

**Strengths:**

1. The experiment results demonstrate great improvements in generative quality, compared to existing differentially private diffusion models.

2. The method reduces the memory footprint by minimizing the batch size requirements and enhances inference efficiency by skipping intermediate sampling steps.

3. The paper is well-structured and clearly written, making it easy to follow the progression of ideas and understand the contributions.

**Weaknesses:**

1. Could the authors further explain the motivation for integrating RAG into DPDM? For example, consider the "public" model (or pre-trained model) which uses public data to train the diffusion model without adding differential privacy noise. Notice that this model does not include sensitive private data. Is it possible that the differential privacy noise introduced in the fine-tuning step will degrade utility, causing RAPID to underperform this public model? What are the possible scenarios where fine-tuning with differential privacy noise can indeed improve utility?

2. See question 2.

**Questions:**

1. What is the possible reason that DPDM improves sharply in the "EMNIST→MNIST" setting as the privacy budget increases, i.e., for FID 125.7 to 12.9 (the best among all methods), as $\varepsilon$ goes from 0.2 to 10?

2. Could the authors further analyze the time complexity of the end-to-end algorithm? Though this method improves inference efficiency by skipping intermediate sampling steps via RAG, is it time-consuming to build the trajectory knowledge base? How does the runtime depend on the knowledge base size? It would be helpful if the authors could provide rough runtime comparisons.

3. If the private data and the public data are too "dissimilar," how will the trajectory knowledge base trained by the public data affect the later sampling steps?

Minor:

4. Line 10 of Algorithm 2: should it be $\tilde{g}$?

---

> ### Author Response · Authors · 2024-11-22
>
> We thank the reviewer for the valuable feedback on improving this paper! Please find below our response to the reviewer’s questions.
>
> > Could the authors further explain the motivation for integrating RAG into DPDM? For example, consider the "public" model (or pre-trained model) which uses public data to train the diffusion model without adding differential privacy noise. Notice that this model does not include sensitive private data. Is it possible that the differential privacy noise introduced in the fine-tuning step will degrade utility, causing RAPID to underperform this public model? What are the possible scenarios where fine-tuning with differential privacy noise can indeed improve utility?
>
> Thanks for the insightful question. In the pre-training/fine-tuning paradigm, public data typically captures diverse patterns while private data is more task-specific and may exhibit distributional differences. Under reasonable privacy constraints, DP fine-tuning using private data tends to improve model performance. For example, under the setting of ImageNet$\rightarrow$CIFAR10, the model pre-trained on the public data (ImageNet) achieves an FID score of 78.63 with respect to the private data (CIFAR10). After DP fine-tuning with RAPID, the score improves to 25.4 at $\epsilon = 10$, although the gain marginally diminishes under stricter privacy budgets (63.2 at $\epsilon = 1$).
>
> > What is the possible reason that DPDM improves sharply in the "EMNIST→MNIST" setting as the privacy budget increases, i.e., for FID 125.7 to 12.9 (the best among all methods), as $\epsilon$ goes from 0.2 to 10?
>
> Unlike pre-training-based approaches, DPDM trains directly on private data with DP, making it more sensitive to privacy constraints. This sensitivity is evident in Table 4, where DPDM's FID scores on CelebA32 deteriorate from 33.0 to 153.1 as $\epsilon$ decreases from 10 to 1. DPDM's strong performance on MNIST at $\epsilon = 10$ can be attributed to the dataset's simplicity (single-channel, grayscale images). For this specific case, using autoencoders and latent diffusion models (as in DP-LDM and RAPID) may be suboptimal. However, DPDM's performance degrades rapidly with stricter privacy budgets or more complex datasets like CIFAR10.
>
> > Could the authors further analyze the time complexity of the end-to-end algorithm? Though this method improves inference efficiency by skipping intermediate sampling steps via RAG, is it time-consuming to build the trajectory knowledge base? How does the runtime depend on the knowledge base size? It would be helpful if the authors could provide rough runtime comparisons.
>
> One key feature of RAPID is its efficient knowledge base generation (detailed in Appendix C5). Unlike prior work on retrieval augmented generation (e.g., ReDi), which requires all the trajectories to share the same latent and builds the knowledge base by iteratively sampling tens of thousands of trajectories from a pre-trained diffusion model, RAPID directly computes the trajectory for each sample in the public dataset in a forward pass. The table below shows RAPID's runtime efficiency for various knowledge base sizes on a workstation running one Nvidia RTX 6000 GPU. For comparison, ReDi requires over 8 hours to build a 10K-sample knowledge base.
>
> |Knowledge Base Size | 10K |         20K |       30K |          40K |         50K |          60K |           70K|
> |:---:|:---:|:---:|:---:|:---:|:---:|:---:|:---:|
> |RAPID|                         2.10s |     4.17s |     6.12s |      8.23s |    10.33s  |   12.19s |     14.48s |

---

> > ### Author Response · Authors · 2024-11-22
> >
> > > If the private data and the public data are too "dissimilar," how will the trajectory knowledge base trained by the public data affect the later sampling steps?
> >
> > To answer this question, we conduct further experiments to evaluate RAPID’s performance when using dissimilar public/private datasets. We use ImageNet32 as the public dataset. For the private dataset, we use VOC2005 (Everingham, 2005), a dataset used for object detection challenges in 2005, which significantly differs from ImageNet32 and contains only about 1K images. We apply RAPID in this challenging setting, with results shown in the table below. Notably, RAPID outperforms baseline methods (e.g., DP-LDM) across various privacy budgets $\epsilon$. While the performance gains are less substantial compared to scenarios with more similar public/private datasets (e.g., ImageNet32$\rightarrow$CIFAR10), the results still demonstrate RAPID's robustness even when working with dissimilar public/private data.
> >
> > | Privacy ($\epsilon$) | DP-LDM | RAPID  |
> > | :-----: | :-----: | :-----: |
> > | 1| 164.85 | 93.17 |
> > |10 |147.86 | 82.56|
> >
> > Moreover, we compare RAPID's performance (without DP) to direct training on the VOC2005 dataset. RAPID improves the FID score from 77.83 to 54.60, showing its ability to effectively leverage the public data even when it differs substantially from the private data.
> >
> > We also explore the possible explanations (details in Appendix D2). Existing studies (Meng et al., 2022; Zhang et al., 2023) establish that in diffusion models, early stages only determine image layouts that can be shared across many generation trajectories, while later steps define specific details. Wu et al. (2023) further discover diffusion models' disentanglement capability, allowing the generation of images with different styles and attributes from the same intermediate sampling stage (as shown in Appendix D2). Therefore, this disentanglement property enables RAPID to maintain robust performance even when public and private dataset distributions differ significantly, provided their high-level layouts remain similar.
> >
> > > Line 10 of Algorithm 2: should it be $\tilde{g}$?
> >
> > Thanks for pointing out this typo, which has been fixed in the revised version.
> >
> > Again, we thank the reviewer for the valuable feedback. Please let us know if there are any other questions or suggestions.
> >
> > Best,
> >
> > Authors

---

> > > ### Comment · Reviewer_iaJi · 2024-11-26
> > >
> > > Thanks to the authors for the reply. I have slightly raised my score.

---

### Official Review · Reviewer_AmQE · 2024-11-04

**Soundness:** 2
**Presentation:** 3
**Contribution:** 3
**Rating:** 5
**Confidence:** 4

**Summary:**

This paper investigates differentially private (DP) data synthesis through diffusion models, proposing an approach that leverages *public* pre-training data to: (1) pre-train diffusion models, (2) train a feature extractor, and (3) derive trajectories from the forward diffusion process to establish a knowledge base. For training on *private* (sensitive) data, this method queries the knowledge base with a noisy *private* sample to retrieve a partially denoised *public* counterpart, which is then used to update the diffusion model via DP-SGD. This process effectively uses public knowledge to bypass certain intermediate training steps, simplifying model fitting. Experimental results indicate promising improvements compared to standard DP diffusion model training.

**Strengths:**

- The paper addresses a relevant and important topic, presenting the material clearly.
- The proposed method is intuitive, straightforward to implement, and readily applicable in practice.
- Experimental results are promising, showing effective improvements over standard DP diffusion model training.

**Weaknesses:**

While the technical contribution is relatively modest (more of a training technique specifically designed for DP diffusion models, which somewhat narrows its scope), the paper’s main value seems to lie in demonstrating performance improvements with this approach.

However, I have concerns regarding the experimental evaluation: (1) some key DP diffusion model baselines are missing, making it difficult to thoroughly assess the claimed performance gains, and (2) the paper lacks adequate investigation into large domain shifts between public and private data, a critical consideration for methods that rely on public data. These gaps leave the reported performance improvements less convincing

**Questions:**

-As previously mentioned, some published works on DP diffusion models are not included in this paper's experimental comparisons, which makes the current evaluation less convincing. Notable examples include:
“Differentially Private Synthetic Data via Foundation Model APIs 1: Images,” ICLR 2024
“PrivImage: Differentially Private Synthetic Image Generation using Diffusion Models with Semantic-Aware Pretraining,” USENIX Security 2024

- Are the timesteps $k$ and $v$ fixed during training? If so, how are these values selected, and what is the robustness to different choices? If not, how is the value of the retrieved timestep $v$ determined, given that the key-value pair only stores the embeddings $(h(x_k), x_v)$?
- For the comparisons, it would improve clarity to present performance across varying $\epsilon$ in a figure, with each method represented as a curve.
- As noted above, the experiments should more thoroughly investigate performance under strong distributional shifts between public and private data (e.g., by using private data with specific domain features, such as medical data).
- As far as I understand, the "Retrieval Augmented Training" approach is orthogonal to model training and architecture. Therefore, it would be more convincing to compare the addition or omission of “Retrieval Augmented Training” on both DPDM and DP-LDM training, showing that adding the proposed  “Retrieval Augmented Training” consistently improves performance in both cases.

---

> ### Author Response · Authors · 2024-11-22
>
> We thank the reviewer for the valuable feedback on improving this paper! Please find below our response to the reviewer’s questions.
>
> > As previously mentioned, some published works on DP diffusion models are not included in this paper's experimental comparisons, which makes the current evaluation less convincing. Notable examples include: “Differentially Private Synthetic Data via Foundation Model APIs 1: Images,” ICLR 2024 “PrivImage: Differentially Private Synthetic Image Generation using Diffusion Models with Semantic-Aware Pretraining,” USENIX Security 2024
>
> We thank the reviewer for highlighting the recent work. We have added a discussion of these papers in Section 2 and conducted comparative experiments with them (details in Appendix C3).
>
> DPSDA (Lin et al., 2024) synthesizes a dataset similar to the private dataset by iteratively querying commercial image generation APIs (e.g., DALL-E 2 and Stable Diffusion) in a DP manner. For a fair comparison with RAPID, instead of using commercial APIs trained on vast datasets (hundreds of millions of images), we train a latent diffusion model (architecture in Appendix B) on ImageNet32 as the query API for DPSDA, using CIFAR10 as the private dataset. The table below shows that RAPID outperforms DPSDA across various $\epsilon$ values in terms of FID scores. Intuitively, RAPID represents a more effective approach for leveraging the public data than querying a generative model trained on such data.
>
> | Privacy ($\epsilon$) | DPSDA| RAPID  |
> | :-----: | :-----: | :-----: |
> | 1| 113.63 | 63.2|
> |10 |60.87 |25.4|
>
> PrivImage (Li et al., 2024) uses a pretraining-then-finetuning approach, querying the private data distribution to select semantically similar public samples for pretraining, followed by DP-SGD finetuning on the private data. The table below compares RAPID and PrivImage's performance across different $\epsilon$ values on CIFAR10 and CelebA64.
>
> CIFAR10
> | Privacy ($\epsilon$) | PrivImage| RAPID  |
> | :-----: | :-----: | :-----: |
> | 1|  29.8 | 63.2|
> |10 |27.6 |25.4|
>
> CelebA64
> | Privacy ($\epsilon$) | PrivImage| RAPID  |
> | :-----: | :-----: | :-----: |
> | 1|  71.4 | 60.5|
> |10 |49.3 |37.3|
>
> RAPID outperforms PrivImage in most scenarios, with one exception: CIFAR10 at $\epsilon = 1$. This likely occurs because PrivImage selects public data similar to the private data for pretraining. With clearly structured private data (for instance, CIFAR10 contains 10 distinct classes), using a targeted subset rather than all the public data tends to improve DP finetuning, especially under strict privacy budgets. However, this advantage may diminish with less structured private data (e.g., CelebA64). We consider leveraging the PrivImage’s selective data approach to enhance RAPID as our ongoing research.
>
> > Are the timesteps $k$ and $v$ fixed during training? If so, how are these values selected, and what is the robustness to different choices? If not, how is the value of the retrieved timestep $v$ determined, given that the key-value pair only stores the embeddings
> $(h(x_k), x_v)?
>
> The parameters $k$ and $v$ remain fixed during training, defining three key phases: initial sampling $(T-k)$ steps, skipping $(k-v)$ steps, and privatization $(v)$ steps. A larger $k$ allows more extensive knowledge base search through longer initial trajectories, enhancing search accuracy at the expense of computational cost. Meanwhile, our ablation study (see Figure 7) shows that increasing $v$ leads to worse FID scores despite better coverage, highlighting a quality-diversity trade-off. Based on empirical testing, we set default values of $k/T = 0.8$ and $v/T = 0.2$ to balance these factors.
>
> > For the comparisons, it would improve clarity to present performance across varying  $\epsilon$ in a figure, with each method represented as a curve.
>
> Following the reviewer’s suggestion, we have included figures to compare the performance of different methods under varying settings of $\epsilon$ (details in Appendix C.6).

---

> > ### Author Response · Authors · 2024-11-22
> >
> > > As noted above, the experiments should more thoroughly investigate performance under strong distributional shifts between public and private data (e.g., by using private data with specific domain features, such as medical data).
> >
> > Thanks for the insightful question! To answer this question, we conduct further experiments to evaluate RAPID’s performance when using dissimilar public/private datasets. We use ImageNet32 as the public dataset. For the private dataset, we use VOC2005 (Everingham, 2005), a dataset used for object detection challenges in 2005, which significantly differs from ImageNet32 and contains only about 1K images. We apply RAPID in this challenging setting, with results shown in the table below. Notably, RAPID outperforms baseline methods (e.g., DP-LDM) across various privacy budgets $\epsilon$. While the performance gains are less substantial compared to scenarios with more similar public/private datasets (e.g., ImageNet32$\rightarrow$CIFAR10), the results still demonstrate RAPID's robustness even when working with dissimilar public/private data.
> >
> > | Privacy ($\epsilon$) | DP-LDM | RAPID  |
> > | :-----: | :-----: | :-----: |
> > | 1| 164.85 | 93.17 |
> > |10 |147.86 | 82.56|
> >
> > Moreover, we compare RAPID's performance (without DP) to direct training on the VOC2005 dataset. RAPID improves the FID score from 77.83 to 54.60, showing its ability to effectively leverage the public data even when it differs substantially from the private data.
> >
> > We also explore the possible explanations (details in Appendix D2). Existing studies (Meng et al., 2022; Zhang et al., 2023) establish that in diffusion models, early stages only determine image layouts that can be shared across many generation trajectories, while later steps define specific details. Wu et al. (2023) further discover diffusion models' disentanglement capability, allowing the generation of images with different styles and attributes from the same intermediate sampling stage (as shown in Appendix D2). Therefore, this disentanglement property enables RAPID to maintain robust performance even when public and private dataset distributions differ significantly, provided their high-level layouts remain similar.
> >
> > > As far as I understand, the "Retrieval Augmented Training" approach is orthogonal to model training and architecture. Therefore, it would be more convincing to compare the addition or omission of “Retrieval Augmented Training” on both DPDM and DP-LDM training, showing that adding the proposed “Retrieval Augmented Training” consistently improves performance in both cases.
> >
> > Thank you for the question. RAPID can integrate with existing methods for training DP diffusion models since it is agnostic to model training, although its neighbor retrieval operates on latents, making it compatible only with latent diffusion models. To measure the impact of RAPID, we use a latent diffusion model as the base model for both DP-LDM and DPDM, evaluating performance with and without RAPID (details in Appendix C4). The table below shows results on MNIST and CIFAR10 at $\epsilon = 10$. The FID score improvement demonstrates the effectiveness of retrieval-augmented training.
> >
> > | | DPDM | |  DP-LDM | |
> > | :-----: | :-----: | :-----: |  :-----: | :-----: |
> > | | w/o | w/ | w/o | w |
> > |MNIST  |42.9| 25.4| 27.2| 14.1 |
> > | CIFAR10 |  82.2 | 54.1 | 33.3 | 25.4 |
> >
> > Again, we thank the reviewer for the valuable feedback. Please let us know if there are any other questions or suggestions.
> >
> > Best,
> >
> > Authors

---

> > > ### Comment · Reviewer_AmQE · 2024-11-25
> > >
> > > Thank you for posting the rebuttal and addressing many of my concerns. While I appreciate the effort, I believe the following points could still be improved:
> > > - Comparison with DPSDA: The comparison would be more convincing if both methods used strong pre-trained models rather than relying on weaker ones. Given that the current presented generation quality is not yet ready for direct application, the experimental results still leave me hesitant about the applicability and performance of the proposed method, which affects my overall assessment of the paper's contribution.
> > > - Inclusion of Additional Methods: Incorporating the newly mentioned baseline methods into the distributional shift experiments and the privacy-utility curves (e.g., Appendix C.6) would provide a more comprehensive evaluation and help address concerns about the claimed superiority of the proposed approach.

---

> > > > ### Author Response · Authors · 2024-11-27
> > > >
> > > > We thank the reviewer's valuable feedback! We have revised the paper to incorporate the reviewer's comments.
> > > >
> > > > > Comparison with DPSDA: The comparison would be more convincing if both methods used strong pre-trained models rather than relying on weaker ones. Given that the current presented generation quality is not yet ready for direct application, the experimental results still leave me hesitant about the applicability and performance of the proposed method, which affects my overall assessment of the paper's contribution.
> > > >
> > > > Notably, RAPID and DPSDA represent two distinct approaches for training DP diffusion models, with the pre-trained model capability affecting their performance differently.
> > > >
> > > > For DPSDA, which uses DP evolution rather than DP training to synthesize data, larger pre-trained models tend to lead to better performance. This is shown in DPSDA's ablation study (Figure 7 in Lin et al., 2024), where increasing the model size from 100M to 270M parameters improves results by enhancing the quality of selected data.
> > > >
> > > > In contrast, methods involving DP training (such as DPDM, DP-LDM, PrivImage, and RAPID) may not benefit from heavily over-parameterized models, as shown by Dockhorn et al., (2023). This is because the $\ell_2$-norm noise added by DP-SGD typically grows linearly with the number of parameters.
> > > >
> > > > To empirically evaluate how the model capability affects two approaches, we conduct experiments varying the size of the pre-trained model from 90M to 337M parameters (by increasing the latent diffusion model's architecture from 128 to 192 channels and its number of residual blocks from 2 to 4). Following the setting in (Li et al., 2024) that replicates DPSDA's results, we use ImageNet32 for pre-training the public model (as the query API for DPSDA) and CIFAR10 as the private dataset (more details in Appendix C3).
> > > >
> > > > | Model Size | 90M | |  337M ||
> > > > | :-----: | :-----: | :-----: | :-----: | :-----: |
> > > > || DPSDA | RAPID | DPSDA | RAPID |
> > > > |      $\epsilon = 1$    |     113.6  |  63.2  |  89.1 |   66.5  |
> > > > |     $\epsilon= 10$    |       60.9 |    25.4     |   43.8   |  29.0|
> > > >
> > > > The table above compares the performance (measured by FID scores) of DPSDA and RAPID across different pre-trained model sizes. As the model complexity increases, DPSDA achieves better FID scores, while RAPID shows marginal performance degradation. Overall, when using the same public dataset and pre-trained model, RAPID consistently outperforms DPSDA, suggesting that it appares more effective at leveraging public data under DP constraints.
> > > >
> > > > > Inclusion of Additional Methods: Incorporating the newly mentioned baseline methods into the distributional shift experiments and the privacy-utility curves (e.g., Appendix C.6) would provide a more comprehensive evaluation and help address concerns about the claimed superiority of the proposed approach.
> > > >
> > > > Following the reviewer’s suggestion, we evaluate DPDSA and PrivImage under the distribution shift setting (details in Appendix C2) and measure their privacy-utility trade-offs (details in Appendix C6).
> > > >
> > > > | Privacy ($\epsilon$) | DP-LDM | DPSDA | PrivImage | RAPID  |
> > > > | :-----: | :-----: | :-----: | :-----: | :-----: |
> > > > | 1| 164.85 |142.20 | 139.07|  93.17 |
> > > > |10 |147.86 | 130.42 | 123.89 | 82.56|
> > > >
> > > > As shown in the table above, under the setting of ImageNet32$\rightarrow$VOC2005, PrivImage demonstrates superior performance compared to DP-LDM and DPSDA, consistent with previously reported results (Li et al., 2024). Moreover, RAPID outperforms all baselines, indicating its effectiveness in leveraging public data even under significant distributional shifts between public and private datasets.
> > > >
> > > > FID Score
> > > > | Privacy ($\epsilon$) | DPDM | DP-LDM | DPSDA | PrivImage | RAPID |
> > > > | :-----: | :-----: | :-----: | :-----: | :-----: | :-----: |
> > > > |      0.2    |     125.7 |   50.8    |   56.3     |    48.8 |    24.0  |
> > > > |           1          |    50.5  |   34.9   |    55.7   |      35.5 |      18.5   |
> > > > |          10        |       12.9 |    27.2     |   52.9   |      27.0 | 14.1 |
> > > >
> > > > Downstream Classification Accuracy
> > > > | Privacy ($\epsilon$) | DPDM | DP-LDM|  DPSDA | PrivImage | RAPID |
> > > > | :-----: | :-----: | :-----: | :-----: | :-----: | :-----: |
> > > > |      0.2    |    85.77%      |  11.35%  | 77.39%  |20.33%  | 96.43%  |
> > > >  |    1    |       95.18%  |     74.62%  | 85.81%  | 85.06% | 98.11% |
> > > >   |       10  |      98.06%   |     95.54% |  90.13% | 91.20% | 99.04% |
> > > >
> > > > The tables above compare the privacy-utility trade-off of different methods under the setting of EMNIST$\rightarrow$MNIST. To ensure a fair comparison, we use EMNIST as public data for model pre-training (as the query API in DPSDA's case) and MNIST as private data. Under equivalent privacy budgets, RAPID consistently outperforms baselines across most scenarios.
> > > >
> > > > Please let us know if you have any additional questions or suggestions.
> > > >
> > > > Best,
> > > >
> > > > Authors

---

### Official Review · Reviewer_EUmh · 2024-11-04

**Soundness:** 3
**Presentation:** 4
**Contribution:** 4
**Rating:** 8
**Confidence:** 3

**Summary:**

This paper introduces RAPID, an approach to training differentially private diffusion models that leverages retrieval augmented generation (RAG). RAPID works by using public data to train a knowledge base of sample trajectories. Then, when training the diffusion model on private data, it retrieves similar trajectories from the knowledge base and focuses on training the later sampling steps in a differentially private manner.

RAPID is compared against baseline private diffusion methods for the task of image synthesis. RAPID outperforms the baselines in most settings in terms of generative quality (measured by FID score).

**Strengths:**

* The paper is clearly written, readable, and well-organized.
* The authors provide a thorough evaluation of RAPID on a variety of benchmark datasets for the task of image synthesis.
* The results show that RAPID outperforms state-of-the-art methods.
* This work opens an avenue for future work incorporating RAG into training.

**Weaknesses:**

- No discussion of the limitation of this approach. See question 2.

**Questions:**

1. Why is the cost of DP hyperparameter tuning small (Line 307)?
2. Does using public data in this way ever reduce the generative quality of the model? For instance, what happens when the public data is wholly unrelated to the private data?

---

> ### Author Response · Authors · 2024-11-22
>
> We thank the reviewer for the valuable feedback on improving this paper! Please find below our response to the reviewer’s questions.
>
> > Why is the cost of DP hyperparameter tuning small (Line 307)?
>
> The general DP hyperparameters follow standard practices, with $\delta$ set smaller than the reciprocal of private data samples. RAPID's unique hyper-parameters are $k$ and $v$, which determine the number of initial sampling steps $(T-k)$, skipped steps $(k-v)$, and privatized steps $(v)$. We explore limited combinations of $k,v$ (e.g., $0.3T$ - $0.7T$). Empirically, RAPID shows robustness to $k/v$ settings, minimizing hyperparameter tuning overhead.
>
> > Does using public data in this way ever reduce the generative quality of the model? For instance, what happens when the public data is wholly unrelated to the private data?
>
> Thanks for the insightful question! To answer this question, we conduct further experiments to evaluate RAPID’s performance when using dissimilar public/private datasets. We use ImageNet32 as the public dataset. For the private dataset, we use VOC2005 (Everingham, 2005), a dataset used for object detection challenges in 2005, which significantly differs from ImageNet32 and contains only about 1K images. We apply RAPID in this challenging setting, with results shown in the table below. Notably, RAPID outperforms baseline methods (e.g., DP-LDM) across various privacy budgets $\epsilon$. While the performance gains are less substantial compared to scenarios with more similar public/private datasets (e.g., ImageNet32$\rightarrow$CIFAR10), the results still demonstrate RAPID's robustness even when working with dissimilar public/private data.
>
> | Privacy ($\epsilon$) | DP-LDM | RAPID  |
> | :-----: | :-----: | :-----: |
> | 1| 164.85 | 93.17 |
> |10 |147.86 | 82.56|
>
> Moreover, we compare RAPID's performance (without DP) to direct training on the VOC2005 dataset. RAPID improves the FID score from 77.83 to 54.60, showing its ability to effectively leverage the public data even when it differs substantially from the private data.
>
> We also explore the possible explanations (details in Appendix D2). Existing studies (Meng et al., 2022; Zhang et al., 2023) establish that in diffusion models, early stages only determine image layouts that can be shared across many generation trajectories, while later steps define specific details. Wu et al. (2023) further discover diffusion models' disentanglement capability, allowing the generation of images with different styles and attributes from the same intermediate sampling stage (as shown in Appendix D2). Therefore, this disentanglement property enables RAPID to maintain robust performance even when public and private dataset distributions differ significantly, provided their high-level layouts remain similar.
>
> Again, we thank the reviewer for the valuable feedback. Please let us know if there are any other questions or suggestions.
>
> Best,
>
> Authors

---

> > ### Comment · Reviewer_EUmh · 2024-11-26
> >
> > Thanks to the authors for updating the submission and addressing my concerns.
> >
> > Regarding appendix C.2, [1], [2], and [3] are some papers that explore dissimilar public/private data in the context of generating private tabular synthetic data. These may improve the discussion. Feel free to include or not.
> >
> > I am keeping my score at 8.
> >
> > [1] Liu, Terrance, et al. "Leveraging public data for practical private query release." International Conference on Machine Learning. PMLR, 2021.
> >
> > [2] Liu, Terrance, et al. "Iterative methods for private synthetic data: Unifying framework and new methods." Advances in Neural Information Processing Systems 34 (2021): 690-702.
> >
> > [3] Fuentes, Miguel, et al. "Joint Selection: Adaptively Incorporating Public Information for Private Synthetic Data." International Conference on Artificial Intelligence and Statistics. PMLR, 2024.

---

> > > ### Author Response · Authors · 2024-11-27
> > >
> > > Thank you for your constructive feedback! We have addressed the limitations of this work and added the relevant references in Section 5.
> > >
> > > Best,
> > >
> > > Authors

---

### Official Review · Reviewer_2WH8 · 2024-11-10

**Soundness:** 3
**Presentation:** 4
**Contribution:** 3
**Rating:** 5
**Confidence:** 3

**Summary:**

The paper presents RAPID, a new approach to train differentially private diffusion models by integrating retrieval-augmented generation (RAG) techniques. RAPID utilizes public data to build a knowledge base of sample trajectories, allowing for efficient use of private data by focusing on training later sampling steps under privacy constraints. This approach improves generative quality, memory efficiency, and inference speed compared to existing methods, demonstrating RAPID's effectiveness in privacy-sensitive generative modeling.

**Strengths:**

Privacy-Utility Trade-off: RAPID effectively balances privacy and utility by using public data for initial steps and focusing on private data for detailed refinement. This selective approach enhances the quality of generated outputs.
Efficiency: The retrieval mechanism reduces memory and computation demands by bypassing intermediate steps, which enhances scalability for larger models.
Generative Quality: Experimental results show that RAPID achieves higher FID and accuracy scores than existing differentially private diffusion models (DPDM), indicating superior output quality and utility.
Innovative Use of Public Data: The approach of leveraging public data to construct a trajectory knowledge base is novel and significantly reduces the computational cost of training on private data.

**Weaknesses:**

1.Over-Reliance on the Quality and Relevance of Public Data. The method’s effectiveness depends heavily on the quality and relevance of the public data used in the early stages of training. If the public data does not adequately represent the private data or is of low quality, the retrieval-augmented generation process will suffer from poor performance.
2.The integration of RAG to enhance the privacy-utility tradeoff could introduce a critical flaw in privacy protection. The retrieval mechanism in RAPID depends on accessing external knowledge bases or data repositories to inform the generative process. This retrieval is based on similar data points or trajectories, which risks the accidental exposure of private data that was used to build the retrieval database.

**Questions:**

Given that the method heavily relies on public data in the early stages, how do you ensure that the public data sufficiently represents or aligns with the private dataset? Have you tested the model’s robustness when using diverse or low-quality public datasets that may not closely match the private data distribution?

---

> ### Author Response · Authors · 2024-11-22
>
> We thank the reviewer for the valuable feedback on improving this paper! Please find below our response to the reviewer’s questions.
>
> > Over-Reliance on the Quality and Relevance of Public Data. The method’s effectiveness depends heavily on the quality and relevance of the public data used in the early stages of training. If the public data does not adequately represent the private data or is of low quality, the retrieval-augmented generation process will suffer from poor performance.
>
> Like other DP diffusion model approaches (e.g., DPDM, DP-LDM, PrivImage) and the broader pre-training/fine-tuning paradigm, RAPID assumes access to a diverse public dataset that captures a range of patterns. However, RAPID is more flexible: the public and private datasets need not closely match in distribution, as long as the public dataset contains similar high-level layouts (more details in Appendix D2).
>
> The explanation is as follows. Existing studies (Meng et al., 2022; Zhang et al., 2023) establish that in diffusion models, early stages only determine image layouts that can be shared across many generation trajectories, while later steps define specific details. Wu et al. (2023) further discover diffusion models' disentanglement capability, allowing generation of images with different styles and attributes from the same intermediate sampling stage (as demonstrated in Appendix D2). Therefore, this disentanglement property enables RAPID to maintain robust performance even when public and private dataset distributions differ significantly, provided their high-level layouts remain similar.
>
> > The integration of RAG to enhance the privacy-utility tradeoff could introduce a critical flaw in privacy protection. The retrieval mechanism in RAPID depends on accessing external knowledge bases or data repositories to inform the generative process. This retrieval is based on similar data points or trajectories, which risks the accidental exposure of private data that was used to build the retrieval database.
>
> We would like to clarify that private data is *only* accessed during model training, not during the sampling process. Meanwhile, the private data's influence on the trained diffusion model is strictly bounded through i) gradient clipping and ii) noise injection (see Algorithm 2), ensuring differential privacy (DP) guarantees. By the post-processing property of DP, since the trained model is differentially private, all its subsequent uses preserve the DP guarantees.
>
> > Given that the method heavily relies on public data in the early stages, how do you ensure that the public data sufficiently represents or aligns with the private dataset? Have you tested the model’s robustness when using diverse or low-quality public datasets that may not closely match the private data distribution?
>
> Following the reviewer’s suggestion, we conduct further experiments to evaluate RAPID’s performance when using degraded and dissimilar public/private datasets. We use ImageNet32 as the public dataset with added Gaussian noise $\mathcal{N}(0, 0.1)$ to degrade its quality. For the private dataset, we use VOC2005 (Everingham, 2005), a dataset used for object detection challenges in 2005, which significantly differs from ImageNet32 and contains only about 1K images. We apply RAPID in this challenging setting, with results shown in the table below. Notably, RAPID outperforms baselines (e.g., DP-LDM) across varying $\epsilon$, indicating its robustness to dissimilar public/private datasets.
>
> | Privacy ($\epsilon$) | DP-LDM | RAPID  |
> | :-----: | :-----: | :-----: |
> | 1| 164.85 | 93.17 |
> |10 |147.86 | 82.56|
>
> Moreover, we compare RAPID's performance (without DP) to direct training on the VOC2005 dataset. RAPID improves the FID score from 77.83 to 54.60, showing its ability to effectively leverage the public data even when it differs substantially from the private data.
>
> We also explore the possible explanations (details in Appendix D2). Existing studies (Meng et al., 2022; Zhang et al., 2023) establish that in diffusion models, early stages only determine image layouts that can be shared across many generation trajectories, while later steps define specific details. Wu et al. (2023) further discover diffusion models' disentanglement capability, allowing the generation of images with different styles and attributes from the same intermediate sampling stage (as shown in Appendix D2). Therefore, this disentanglement property enables RAPID to maintain robust performance even when public and private dataset distributions differ significantly, provided their high-level layouts remain similar.
>
> Again, we thank the reviewer for the valuable feedback. Please let us know if there are any other questions or suggestions.
>
> Best,
>
> Authors

---

### Meta-Review · Area_Chair_dn1H · 2024-12-20

**Metareview:**

The paper presents a retrieval augmented training approach for DP diffusion models by using a database of intermediate diffusion steps from public data to shortcut diffusion simulation and make it more efficient.

**Strengths:**
* Impressive empirical performance on highly relevant problem
* Clear presentation
* After rebuttal, comprehensive evaluation of baselines and ablations

**Weaknesses:**
* Reliance on relevant public data. While the authors added experiments to attempt to assess performance when public data is not as good, more comprehensive and stronger evaluation would be useful

Overall, the paper makes a nice contribution to improve DP diffusion models, displaying impressive empirical benefits across a range of scenarios. The main weakness of reliance on relevant public data is a feature of this genre of methods and not specific to the current submission, although more comprehensive evaluation to understand the breaking point would be interesting. Nevertheless, the strengths clearly outweigh this weakness and the paper should be accepted.

**Additional Comments On Reviewer Discussion:**

After the end of discussion phase on OpenReview, Reviewer AmQE confirmed by email that their concerns had been addressed.

Reviewer 2WH8 never responded to the author rebuttal, but based on my evaluation the authors were able to answer their concerns.

---

### Decision · Program_Chairs · 2025-01-22

Accept (Poster)